# Characterization of the Complete Mitogenomes of Four Dacinae Species (Diptera: Tephritidae) with Phylogenetic Analysis

**DOI:** 10.3390/ani15223301

**Published:** 2025-11-15

**Authors:** Deliang Xu, Shuangmei Ding, Xiaojie Zeng, Lele Du

**Affiliations:** 1Key Laboratory of Applied Ecology on the Loess Plateau, Shaanxi Province, School of Life Science, Yan’an University, Yan’an 716000, China; 15229158172@163.com (X.Z.); dulele1128@163.com (L.D.); 2The Institute of Scientific and Technical Research on Archives, National Archives Administration of China, Beijing 100050, China; shuangmeiding@163.com

**Keywords:** true fruit flies, mitochondrial genome, PhyloBayes, Gastrozonini

## Abstract

Dacinae comprises over 1400 species and 51 genera divided into three tribes, including Ceratitidini, Dacini, and Gastrozonini. Dacine fruit flies are frugivorous insects that represent a diverse and economically significant group, widely recognized as a notorious major agricultural pest. Mitochondrial genomic data provide powerful tools for elucidating the phylogenetic relationships within Dacinae and offering deeper insights into the evolutionary history of its major lineages. In this study, we presented the complete mitochondrial genomes of four species including *Acroceratitis separata*, *Acrotaeniostola quadrivittata*, *Gastrozona parviseta*, and *Paragastrozona vulgaris* from the tribe Gastrozonini. These mitogenomes displayed conserved structural and compositional features consistent with those observed in other fully sequenced dacine genomes. We reconstructed the phylogenetic relationships within Dacinae, indicating robust support for the monophyly of this group. This research substantially expands the available mitogenomic data, thereby enhancing our comprehension of the evolutionary trajectories and phylogenetic structure within the Dacinae.

## 1. Introduction

The subfamily Dacinae (Diptera: Tephritidae), commonly known as fruit flies, are frugivorous insects that constitute a diverse and economically significant group, recognized as a notorious major agricultural pest. This subfamily of more than 1400 species within 51 genera comprises three tribes: the Ceratitidini (20 genera, 198 species), Dacini (4 genera, 932 species), and Gastrozonini (27 genera, 137 species) [1,2,3,4,5,6], with representatives found in the Old World tropical, subtropical, and warm temperate areas, except for a few species that have been introduced into many parts of the New World. The members of this subfamily are characterized by postocular setae all slender, pointed, dark brown to black; scapular setae present; vertical suture of anepisternum well developed; scutellum generally swollen, convex dorsally; vein R_4+5_ setose at least extending to r-m cross-vein; lobe of basal cubital cell (bcu) conspicuously prolonged; female with only two spermathecae. Dacine fruit flies are naturally polyphagous, displaying high reproduction potential, extensive climatic tolerance, and significant dispersal ability [7]. The larvae of most species of Ceratitidini and Dacini are fruit-feeding, infesting the fruits of a wide range of host plants, although a few dacine species breed in flowers [8,9,10]. Whereas those of the Gastrozonini develop primarily in the bamboo shoots, a few species of those whose biology is known tend to attack living stems of Poaceae [11,12,13].

Despite extensive research into the evolutionary history of Dacinae’s subfamily, and a classification of tribes and genera, along with species composition, a consensus concerning their taxonomy and phylogeny continues to be challenged by ongoing scholarly debates. Pioneering studies within this subfamily were undertaken by describing new regional species and only partially addressing the required taxonomic revisions [14,15,16]. Norrbom et al. (1999) deemed this group to be the tribe Dacini, including three subtribes: Gastrozonina, Ceratitidina and Dacina [17], while Korneyev (1999) recognized the Ceratitidini, Dacini, and Gastrozonini in the subfamily Dacinae [1]. But these two tribes Ceratitidini and Gastrozonini together formed the Ceratitidinae, treating them as a separate subfamily [18,19,20]. Recently, phylogenetic studies have confirmed the monophyly of Dacinae based on morphological characteristics [21], as well as nuclear and mitochondrial gene fragments [22,23,24]. Correspondingly, the Dacini and Ceratitidini were recovered as monophyletic, but the Gastrozonini was considered to possibly be paraphyletic [1,17]. Furthermore, Doorenweerd et al. (2018) provided a much-needed and global checklist of valid species names for the Dacini, contributing to the resources for studying fruit flies and components of their impact on agriculture [4]. Phylogenomic analyses employing the universal single-copy ortholog (BUSCO), ultraconserved element (UCE), and anchored hybrid enrichment (AHE) datasets reaffirmed the monophyly of Dacinae [25]. Basically, morphological and molecular frameworks for the phylogenetic relationships among major dacine lineages have not yet been satisfactorily defined. Whatever the nominal rank of these tribes and genera is, higher-level phylogenetic analyses are necessary, and their classification status is still unresolved. Consequently, to fully elucidate the relationships within Dacinae, more detailed phylogenetic studies incorporating broader taxonomic sampling are needed, while mitochondrial genomic data can offer valuable insights into reconstructing the phylogeny of Dacinae and further clarifying the relationships among its major lineages.

Insect mitochondrial genomes (mitogenomes) serve as powerful molecular markers providing a wealth of genetic information. Particularly in the multiple protein-coding genes (PCGs), rRNA genes and tRNA genes significantly increase the genomic information available for phylogenetic analyses, enhancing the resolution of phylogenetic trees. Mitogenome genes can offer insights into the divergence and evolution within the different lineages. Furthermore, its base compositional heterogeneity, accelerated evolutionary rate, stable composition, relatively conservative structure, small size (14–17 kb), and strict maternal inheritance make it invaluable for investigating genome evolution, population genetic structure, species identification, and biogeography in insects [26,27,28,29]. Within the Dacinae, mitogenome data has been widely used in many studies to investigate the relationships among major lineages across different taxonomic levels [30,31,32,33,34,35], bolstering our understanding of mitogenome diversity within this group. However, phylogenetic analyses have yielded inconsistent results showing tribal relationships varying from (Ceratitidini + (Dacini + Gastrozonini)) to (Dacini + (Ceratitidini + Gastrozonini)) based on different taxon samplings, datasets, analytical methods, and models. At present, only one complete mitochondrial genome concerning the bamboo-shoot fruit fly *Acrotaeniostola dissimilis* has been sequenced for Gastrozonini [36]. Comprehensive comparative mitogenome studies have not been conducted for Dacinae, and expanding mitogenomic data could enable deeper analyses to clarify tribal-level phylogenetic relationships within this subfamily.

To advance Dacinae phylogenetics and evolution, we sequenced and analyzed the complete mitogenomes of four species (*Acroceratitis separata*, *Acrotaeniostola quadrivittata*, *Gastrozona parviseta*, and *Paragastrozona vulgaris*) from the tribe Gastrozonini for the first time. We characterized the general features of these four new mitogenomes and performed comprehensive comparative mitogenome analyses across 46 Dacinae species. Leveraging this extensive and unique taxon sampling, we systematically assessed the efficacy of site-homogeneous models and site-heterogeneous models to resolve the Dacinae phylogenetic relationships. This study significantly enriches the available mitogenomic data for Dacinae, thereby providing robust support for phylogenetic investigations and unlocking novel insights into this group’s evolutionary history and higher-level relationships.

## 2. Materials and Methods

### 2.1. Sample Collection and DNA Isolation

Specimens of *Acroceratitis separata* were collected from Shiqian Town, Shixing County, Shaoguan City, Guangdong Province, 26 July 2023, collected by Deliang Xu; *Acrotaeniostola quadrivittata* and *Gastrozona parviseta* were captured from the Wangtianshu Scenic Area, Mengla County, Yunnan Province, 22 April 2024, collected by Sheng Gao; while *Paragastrozona vulgaris* was collected from the Daguan County, Zhaotong City, Yunnan Province, July 2024, Malaise trap. Four species of Dacinae were identified by external morphology and genitalia. All samples were kept in 100% ethyl alcohol and stored at −20 °C. Voucher specimens were deposited in the Insect Collection, School of Life Sciences, Yan’an University, Yan’an, China.

Total genomic DNA was extracted from the abdomen tissues of adult specimens using the DNA Kit (TransGen Biotech, Beijing, China) following the manufacturer’s protocols.

### 2.2. Mitogenome Sequencing, Assembly and Annotation

The mitogenome sequences were obtained using next-generation sequencing at the Illumina Xplus platform with 150 bp paired-end reads (Berry Genomics, Beijing, China). The raw sequencing data were preprocessed to filter adapters and low-quality reads, obtaining high-quality clean data for subsequent genome assembly and annotation using the Geneious Prime^®^ 2023.2.1 [37], with the mitochondrial genome of *Acrotaeniostola dissimilis* (MH900079) being selected as reference sequences [36].

All 13 PCGs were determined by their open reading frames based on the invertebrate mitochondrial genetic code and alignment with homologous genes of other true fruit flies. The 13 PCGs were manually annotated using start codons (ATN, GTG, TTG, CAA) and stop codons (T−, TAA, TAG). The 22 tRNAs and their secondary structures were annotated using the MITOS2 (https://usegalaxy.eu/; accessed on 8 November 2025) [38]. Two rRNAs were predicted by alignment with other reference mitogenomes and their boundaries were defined by the adjacent tRNAs. Moreover, circular maps of the complete mitochondrial genome were produced using the Proksee website (https://proksee.ca/; accessed on 8 November 2025) [39]. The four newly annotated mitogenome sequences were submitted to GenBank and given the accession numbers PV920620 (*Acroceratitis separata*), PV920621 (*Acrotaeniostola quadrivittata*), PV920622 (*Gastrozona parviseta*), and PV920623 (*Paragastrozona vulgaris*) (Table 1).

### 2.3. Comparative Mitochondrial Genomic Analysis

Nucleotide composition skew was computed according to the following formulas: AT-skew = (A − T)/(A + T) and GC-skew = (G − C)/(G + C). The organization of the mitogenomes, codon usage of PCGs, nucleotide content, and nucleotide skew of these four species were examined using PhyloSuite v1.2.3 [40]. A total of 46 mitogenomes representing 3 tribes were utilized to carry out comparative mitochondrial genome analyses in order to explore the mitogenome nucleotide differences and gene evolution within Dacinae. The size of the mitogenome, AT content, and nucleotide composition skew were analyzed throughout the entire mitochondrial genome as well as within other components including PCGs, RNAs, and CR. Subsequently, nucleotide similarity comparisons of PCGs between *Acroceratitis separata* and remaining Dacinae mitogenomes were conducted using BLASTP searches in the CGView Comparison Tool V3 (CCT) (https://github.com/paulstothard/cgview_comparison_tool; accessed on 8 November 2025) [41]. Eventually, the AT content for each component of the Dacinae mitogenomes and a scatter plot of the full-length sequences were visualized using GraphPad Prism V9.0.0 (San Diego, CA, USA, https://www.graphpad.com). OriginPro 2024 (Northampton, MA, USA, https://www.originlab.com) was utilized to generate three-dimensional scatter plots that incorporated the AT-skew, GC-skew, and AT content of the 46 mitochondrial genomes.

### 2.4. Indices of Codon Usage Bias

Codon usage bias (CUB) is an important genomic characteristic that describes the uneven utilization of synonymous codons in the PCGs, preferentially employing some of the specific codons during the evolutionary processes. The assessment indicators for CUB encompass the relative synonymous codon usages (RSCU), the effective number of codons (ENC), the combination of ENC and GC content of synonymous codons at the 3rd codon position (ENC-GC3s plot), the comparison between the average of GC content at 1st and 2nd codon position, and the GC content at 3rd codon position (GC12-GC3, neutrality plot), as well as the corresponding A, T, G, and C content at their synonymous 3rd codon position (A3s/(A3s + T3s) − G3s/(G3s + C3s), Parity Rule 2 bias plot).

The RSCU values for each PCG in the mitochondrial genome were calculated using PhyloSuite v1.2.3 [40]. A heatmap was generated using TBtools-II v2.096 [42] based on amino acid (AA) counts, AA ratios, and total codon counts across the entire mitogenomes, employing a logarithmic scale and normalization. Typically, an ENC threshold of 35 is regarded as the boundary for assessing the degree of codon preference. An ENC value ≤ 35 indicates a strong codon preference, while a value exceeding 35 suggests a weaker preference. Additionally, we employed the standard simulated curve under ideal conditions and calculated the method: ENC = 2 + GC3s + 29/(GC3s^2^ + (1 − GC3s)^2^). We can determine if mutation plays a dominant role in codon preference by observing the placement of the ENC-GC3s point on or near the standard curve. Conversely, if the points fall below the standard curve, it indicates that the selection is the primary influence on codon preference. Finally, ENC and GC3s values were obtained using CodonW v.1.4.2 [43] and the curve graph was generated using GraphPad Prism V9.0.0.

The neutrality plot analyzes the correlation between GC12 on the y-axis and GC3 on the *x*-axis. A significant correlation with points along a diagonal near the standard curve and a regression coefficient close to 1 suggests that mutation pressure mainly influences codon preference. In contrast, no or very low correlation with a nearly flat regression curve on the *x*-axis and a regression coefficient below 0.5 indicates that natural selection dominates codon preference. Eventually, GC12-GC3 values were calculated using PhyloSuite v1.2.3 [40] and a regression curve was generated with GraphPad Prism V9.0.0.

An analysis of PR2 bias was conducted to evaluate the impact of natural selection and mutational pressure on codon bias, employing the formulas A3s/(A3s + T3s) as the ordinate and G3s/(G3s + C3s) as the abscissa. Theoretically, these data points fall into four quadrants. The central point, which is equivalent to 0.5, indicates that the influence of selection and mutation is equal. Nevertheless, analyzing deviations in AT and GC pairs provides more precise information regarding the strength of biases due to neutral mutations. Lastly, PR2 bias values were derived using CodonW v.1.4.2 [43] and scatter diagrams were plotted using GraphPad Prism V9.0.0.

### 2.5. Sequence Alignment and Analysis of Sequence Heterogeneity

A total of 52 mitogenomes were utilized for multiple alignment and phylogenetic analysis, comprising 46 mitogenomes from three tribes within the Dacinae subfamily, incorporating four newly sequenced specimens and 42 publicly available mitogenomes, which formed the ingroup for the study. Six species from four different subfamilies were designated as outgroup taxa (Table 1). For subsequent analyses, 46 mitogenomes encompassing all PCGs, AA, and rRNAs were extracted with PhyloSuite v1.2.3 [40]. Each PCG was individually aligned using codon-based multiple alignments mode with the G-INS-i strategy under the MAFFT algorithm, as implemented in the PhyloSuite v1.2.3 [40]. AA sequences underwent a similar alignment process. For all the rRNAs, separate alignments were performed using the MAFFT v7 online service (https://mafft.cbrc.jp/alignment/server/; accessed on 8 November 2025) [44] employing the Q-INS-i algorithm. Unreliable and ambiguous sites in the alignments were then eliminated by Gblocks 0.91b [45]. Finally, multiple alignment sequences were concatenated using PhyloSuite v1.2.3 [40] to generate five distinct datasets: (1) a PCG123 matrix containing all three codon positions of 13 PCGs (11,136 bp); (2) a PCG123R matrix combining all three codon positions of 13 PCGs with two rRNAs (13,010 bp); (3) a PCG12 matrix, with only the first and second codon positions of 13 PCGs (7424 bp); (4) a PCG12R matrix, comprising the first and second codon positions of 13 PCGs plus two rRNAs (9298 bp); (5) and an AA matrix, representing translated amino acid sequences from 13 PCGs (3706 bp).

To assess heterogeneous sequence divergence across five alignment datasets, AliGROOVE v1.05 [46] was employed with default sliding-window settings. DNA indels in nucleotide datasets were treated as ambiguous and the BLOSUM62 matrix served as the default for amino acid substitutions. This metric facilitates pairwise sequence comparisons of individual terminals or subclades with terminals outside of the focal group. The scoring distances between sequences were compared for similarity within the whole data matrix.

**Table 1 animals-15-03301-t001:** Mitochondrial genomes used for phylogenetic analysis in the present study.

Subfamily	Tribe	Species	Accession Number	Reference
Dacinae	Ceratitidini	*Ceratitis (Ceratalaspis) cosyra*	MT036784	[47]
Dacinae	Ceratitidini	*Ceratitis (Ceratalaspis) pallidula*	MT036775	[47]
Dacinae	Ceratitidini	*Ceratitis (Ceratalaspis) quinaria*	MT036788	[47]
Dacinae	Ceratitidini	*Ceratitis (Ceratitis) capitata*	NC_000857	[48]
Dacinae	Ceratitidini	*Ceratitis (Pardalaspis) bremii*	MT036781	[47]
Dacinae	Ceratitidini	*Ceratitis (Pardalaspis) ditissima*	MT036785	[47]
Dacinae	Ceratitidini	*Ceratitis (Pardalaspis) punctata*	MT036786	[47]
Dacinae	Ceratitidini	*Ceratitis (Pterandrus) anonae*	MT036768	[47]
Dacinae	Ceratitidini	*Ceratitis (Pterandrus) quilicii*	MT036780	[47]
Dacinae	Ceratitidini	*Ceratitis (Pterandrus) rubivora*	MT036789	[47]
Dacinae	Ceratitidini	*Neoceratitis asiatica*	MF434829	[31]
Dacinae	Dacini	*Bactrocera (Bactrocera) carambolae*	MT121283	[49]
Dacinae	Dacini	*Bactrocera (Bactrocera) correcta*	NC_018787	[50]
Dacinae	Dacini	*Bactrocera (Bactrocera) dorsalis*	NC_008748	[51]
Dacinae	Dacini	*Bactrocera (Bactrocera) propinqua*	OR085850	[52]
Dacinae	Dacini	*Bactrocera (Bactrocera) thailandica*	MT121271	[49]
Dacinae	Dacini	*Bactrocera (Bactrocera) wuzhishana*	NC_071744	[49]
Dacinae	Dacini	*Bactrocera (Daculus) biguttula*	NC_042712	[53]
Dacinae	Dacini	*Bactrocera (Daculus) oleae*	PQ801851	[54]
Dacinae	Dacini	*Bactrocera (Tetradacus) minax*	NC_014402	[55]
Dacinae	Dacini	*Bactrocera (Tetradacus) tsuneonis*	NC_038164	[56]
Dacinae	Dacini	*Dacus (Callantra) longicornis*	NC_032690	[57]
Dacinae	Dacini	*Dacus (Dacus) bivittatus*	NC_046468	[58]
Dacinae	Dacini	*Dacus (Dacus) durbanensis*	NC_071727	[49]
Dacinae	Dacini	*Dacus (Dacus) eclipsis*	NC_071728	[49]
Dacinae	Dacini	*Dacus (Dacus) venetatus*	NC_071730	[49]
Dacinae	Dacini	*Dacus (Didacus) ciliatus*	MG962405	[58]
Dacinae	Dacini	*Dacus (Didacus) humeralis*	NC_071729	[49]
Dacinae	Dacini	*Dacus (Mellesis) conopsoides*	NC_043843	[33]
Dacinae	Dacini	*Dacus (Mellesis) trimacula*	MK940811	[59]
Dacinae	Dacini	*Dacus (Mellesis) vijaysegarani*	MW429439	[34]
Dacinae	Dacini	*Zeugodacus (Parasinodacus) cilifer*	NC_052852	Direct submission
Dacinae	Dacini	*Zeugodacus (Sinodacus) hochii*	NC_071734	[49]
Dacinae	Dacini	*Zeugodacus (Sinodacus) triangularis*	NC_071736	[49]
Dacinae	Dacini	*Zeugodacus (Zeugodacus) caudatus*	NC_062801	[60]
Dacinae	Dacini	*Zeugodacus (Zeugodacus) cucurbitae*	OQ158899	Direct submission
Dacinae	Dacini	*Zeugodacus (Zeugodacus) depressus*	MT477832	Direct submission
Dacinae	Dacini	*Zeugodacus (Zeugodacus) diaphorus*	NC_028347	[30]
Dacinae	Dacini	*Zeugodacus (javadacus) calumniatus*	NC_079944	[35]
Dacinae	Dacini	*Zeugodacus (javadacus) heinrichi*	NC_079945	[35]
Dacinae	Dacini	*Zeugodacus (javadacus) tau*	MH900081	[61]
Dacinae	Gastrozonini	*Acroceratitis separata*	PV920620	This study
Dacinae	Gastrozonini	*Acrotaeniostola dissimilis*	MH900079	[36]
Dacinae	Gastrozonini	*Acrotaeniostola quadrivittata*	PV920621	This study
Dacinae	Gastrozonini	*Gastrozona parviseta*	PV920622	This study
Dacinae	Gastrozonini	*Paragastrozona vulgaris*	PV920623	This study
Trypetinae	Carpomyini	*Carpomya vesuviana*	NC_071721	[49]
Tachiniscinae	Ortalotrypetini	*Cyaforma shenonica*	PV866838	Unpublished
Phytalmiinae	Acanthonevrini	*Felderimyia fuscipennis*	NC_052851	Direct submission
Trypetinae	Carpomyini	*Myiopardalis pardalina*	NC_087865	Direct submission
Tephritinae	Cecidocharini	*Procecidochares utilis*	NC_020463	Direct submission
Tephritinae	Tephritini	*Tephritis femoralis*	NC_047184	[62]

### 2.6. Phylogenetic Analysis

Phylogenetic analyses for the five datasets were conducted using both Bayesian inference (BI) and Maximum Likelihood (ML) methods with the site-homogeneous models. The optimal partitioning schemes and base substitution models for BI and ML were determined through PartitionFinder 2 [63], employing the BIC criterion and a greedy algorithm (Appendix A). The BI analysis was performed with MrBayes 3.2.6 [64] on the CIPRES Science Gateway (https://www.phylo.org; accessed on 8 November 2025), utilizing two independent runs spanning 6–30 million generations, sampling every 1000 generations, and discarding the initial 25% as burn-in. Concurrently, for the ML analysis, IQ-TREE v.1.6.8 [65] was employed with the ultrafast bootstrap (UFB) algorithm, conducting 1000 replicates. Additionally, Bayesian analysis on all datasets was carried out using PhyloBayes MPI v1.8c [66] with the site-heterogeneous mixture model CAT + GTR, also on the CIPRES Science Gateway (https://www.phylo.org; accessed on 8 November 2025). Two independent chains were initiated from random topologies and continued until convergence was achieved when the maxdiff < 0.3 and minimum effective size > 50. A consensus tree was generated by taking a tree every ten cycles from the remaining trees after discarding the initial 25% as burn-in. All phylogenetic trees were visualized and refined using the iTOL v6 [67] and Adobe Illustrator CC 2018.

## 3. Results

### 3.1. The General Features of Mitogenome Organization

The complete mitochondrial genomes of four representative species within the Dacinae are characterized by closed, circular, and double-stranded DNA molecules, comprising 37 mitochondrial genes (13 PCGs, 22 tRNA genes, and two rRNA genes) and an A+T-control region, exhibiting notable size variation ranging from 16,112 bp to 16,691 bp. Specifically, *Acroceratitis separata*, *Acrotaeniostola quadrivittata*, *Gastrozona parviseta*, and *Paragastrozona vulgaris* possessed mitochondrial genome lengths of 16,603 bp, 16,112 bp, 16,691 bp, and 16,594 bp, respectively (Figure 1; Appendix A). These four newly sequenced mitochondrial genomes, in conjunction with other publicly available dacine mitochondrial sequences retrieved from GenBank, were utilized to perform a comprehensive comparative analysis of mitochondrial genomes within the family Dacinae. As illustrated in the scatterplot depicted in Figure 2, which displayed the mitochondrial genome sizes of 46 Dacinae species, *Zeugodacus* (*Zeugodacus*) *caudatus* had the smallest mitogenome at 15,311 bp, whereas *G. parviseta* exhibited the largest at 16,691 bp. The observed size variation among Dacinae mitogenomes primarily correlated with fluctuations in the length of the A+T-rich region, which remained highly mutable and contributed remarkably to genome size heterogeneity within this subfamily. Additionally, the gene order observed in sequenced dacine mitogenomes was highly conserved and largely aligned with the insect ancestral mitochondrial arrangement, with the exception of tRNA gene duplication in *Dacus* (*Mellesis*) *conopsoides*.

### 3.2. Comparative Analysis of Nucleotide Composition

The nucleotide compositions of *A. separata*, *A. quadrivittata*, *G. parviseta*, and *P. vulgaris* revealed a notably high AT content across their entire mitochondrial genomes, with values reaching 77.4%, 78.4%, 75.1%, and 75.1%, respectively. This pronounced AT bias was consistently observed across multiple genomic regions, including PCGs, RNAs, and CR, whereas the AT content in the control region peaked at 82.1% in *A. separata* (Appendix A). The skew statistics for these four species indicated a preference for A and C throughout whole genomes, characterized by a positive AT-skew and a negative GC-skew (Appendix A). Furthermore, the 46 mitogenomes of Dacinae exhibited a pronounced AT bias, with AT content ranging from 66.6% in *Bactrocera* (*Tetradacus*) *tsuneonis* (Dacini) to 79% in *Neoceratitis asiatica* (Ceratitidini) (Figure 3). The average AT content across whole Dacinae mitogenomes was approximately 74.14%, which was significantly higher than that observed in the average AT content of PCGs but lower than that of the average AT content of rRNAs, tRNAs, and the A+T-control region (Figure 4). Accordingly, consistent with the observed trend in overall AT content, PCGs, RNAs, and CR all demonstrated elevated levels of AT richness, reflecting a conserved compositional bias across these genomic components. Moreover, analysis of the AT-skew across the entire mitochondrial genome revealed predominantly positive values, ranging from 0.008 in Gastrozonini, *A. separata* to 0.15 in Dacini, *B*. (*T*.) *tsuneonis*. Conversely, GC-skew values were mainly negative, varying from −0.34 in Dacini, *B*. (*T*.) *tsuneonis* to −0.153 in Gastrozonini, *A. quadrivittata* (Figure 3). In the PCG and RNA regions, negative AT-skew values and mostly positive GC-skew values were consistently observed. Notably, the magnitude and variability of both AT-skew and GC-skew were highly pronounced within the Dacinae. Explicitly, these findings demonstrated a pronounced nucleotide bias toward A and C bases within the mitochondrial genomes of Dacinae species, while there was a discernible preference for T and G skew in PCGs and RNAs.

### 3.3. PCGs and CCT Analysis

The combined length of PCGs in *Acroceratitis separata*, *Acrotaeniostola quadrivittata*, *Gastrozona parviseta*, and *Paragastrozona vulgaris* was determined to be 11,181 bp, 11,175 bp, 11,172 bp, and 11,172 bp, respectively. These genomes exhibited a notably high AT content, accounting for 76.1%, 77.4%, 73.6%, and 74.2% (Appendix A). Further analysis revealed that the third codon position displayed the highest AT richness, measuring 93.6%, 95.1%, 87.9%, and 89.8%, in comparison to the first codon position, which had AT contents of 68.7%, 70.2%, 67%, and 66.8%, and the second codon position, which recorded values of 66%, 66.9%, 65.8%, and 65.8% across the four species (Appendix A). Notably, the first codon position exhibited a bias favoring TG nucleotides, whereas the second and third positions showed a skew towards TC nucleotides. Among the 46 sequenced species, the third codon position of PCGs exhibited the highest AT content when compared to the average values of the corresponding components across all species except for CR, with values ranging remarkably from 65.6% in Dacini species *B*. (*T*.) *tsuneonis* to 95.1% in the Ceratitidini species *N. asiatica* and Gastrozonini species *A. quadrivittata*, with an overall average of 84.27% (Figure 4). By contrast, the second codon position consistently demonstrated the lowest AT content among the three codon positions, with values spanning from 63.2%, observed in Dacini species (*B*. (*T*.) *tsuneonis*), to 67.2% in Ceratitidini species (*N. asiatica*), culminating in an average of 65.52%. Meanwhile, in the nucleotide composition analysis among all PCGs, the *ND4L* exhibited the highest average AT content, reaching approximately 78.78%, whereas the *COI* displayed the lowest at 65.86% (Figure 4). Furthermore, after conducting a comprehensive analysis of the PCGs within the 46 dacine mitogenomes, we observed that the majority of PCGs predominantly utilized the canonical ATN as the start codons, namely ATA, ATT, ATG, and ATC. However, notable exceptions were identified in the *COI*, *ATP8*, *ND1*, and *ND5*, which employed atypical initiation codons such as CAA, TCG, TTG, and GTG in certain species. Correspondingly, all PCGs mainly concluded with either TAA or TAG, with TAA being the most frequent. Nevertheless, the incomplete stop codon T– was detected in some genes.

We utilized the complete PCGs of *A. separata* as a reference sequence to perform pairwise comparison analyses across other Dacinae mitochondrial genomes. The comparative results derived from BLAST + 2.17.0 analyses and circular genome mappings were presented through the CCT (Figure 5). Sequence similarity was observed at the PCG level between *A. separata* and other Dacinae mitogenomes, varying from 78.1% (*A. Separata* vs. *B.* (*T*.) *tsuneonis*) to 90.15% (*A. Separata* vs. *Acrotaeniostola dissimilis*), indicating a notable degree of consistency in the PCGs across these species. In addition, *A. separata* demonstrated the closest relationship with *A. dissimilis* and *A. quadrivittata* (sequence similarities of approximately 89.63%), followed by *P. vulgaris* at 88.84% and *G. parviseta* at 88.49%. These findings supported the inference that these species were phylogenetically related based on PCG level. Nevertheless, the observed similarities in protein level among these species did not entirely correspond with the anticipated phylogenetic relationships (Appendix A). Moreover, the similarities among the *ND2*, *ATP8*, *ND3*, *ND5*, and *ND6* were relatively low, indicating that these genes were sequentially variable and less conserved. In contrast, the *COI*, *COIII*, and *CYTB* were characterized by a relatively lower degree of variation and were relatively more conserved than other genes (Figure 5).

### 3.4. Comparative Analysis of CUB

The RSCU values for four Gastrozonini mitogenomes were computed and presented in Figure 6. The analysis revealed that Leu2 (UUA), Ile (AUU), Phe (UUU), and Gly (GGA) were the most prevalent utilized amino acids, reflecting a pronounced bias toward nucleotides A and U. This trend aligned with the overall high AT content observed in the PCGs of Gastrozonini species. Furthermore, we carried out hierarchical clustering analysis for the heatmaps related to three aspects: the AA usage ratio across individual species (Figure 7), along with corresponding AA usage count (Appendix A), and total codon usage counts (Appendix A). The heatmap visualization clearly illustrated differences in the AA usage ratio and count, as well as codon usage count, with a gradient ranging from blue (low values) to red (high values). Totally, these codon bias patterns were reaffirmed, yielding consistent results.

The observed ENC and ENC-GC3s values were derived from the complete 13 PCGs across 46 mitochondrial genomes, each individual PCG of 46 mitogenomes species within the Dacinae, and the 13 PCGs from four Gastrozonini mitogenomes, respectively (Figure 8). The results revealed that the ENC values exhibited considerable variability among the individual PCGs, ranging from 23.96 to 58.91. This range indicated the presence of both strong and weak CUB across different mitochondrial genome species. Simultaneously, our ENC-GC3s plot analysis revealed that the *ATP8*, *ND4L*, and *ND6* in certain species were notably positioned above the expected curve, indicating that mutation served as the predominant factor influencing codon usage preference. Conversely, in the remaining PCGs, as well as each individual PCG across the 46 Dacinae mitogenomes, codon usage bias appeared to be predominantly shaped by selection. Additionally, four Gastrozonini mitogenomes demonstrated that natural selection exerted a more significant influence on codon usage than mutation. Collectively, our results supported the hypothesis that both mutation and selection collaboratively contributed to the evolutionary dynamics of codon usage in Dacinae mitochondrial genomes.

Based on the neutral plot analysis, regression coefficients below 0.5 imply that natural selection predominantly influences CUB across all 13 PCGs in 46 mitogenomes and each individual PCG within these species (Figure 9). In contrast, *A. separata*, *G. parviseta*, and *P. vulgaris* exhibited regression coefficients approaching or exceeding 1, signifying a robust correlation between GC12 and GC3, suggesting that the variation in base composition among the three codon positions was minimal within these species. In contrast, *A. quadrivittata* was found to be below 0.5, indicating a distinct preference for the base composition at the codon positions. As a result, this pattern suggested that natural selection exerted a more significant impact on codon usage bias, with mutation pressure playing a comparatively minor role in these Dacinae mitochondrial genomes.

We further analyzed the correlation between the frequency of AT and the GC in the 13 PCGs using the PR2 bias plot (Figure 10). Our results demonstrated that the majority of genes clustered within the second quadrant across the entire 13 PCGs of 46 Dacinae mitogenomes. This spatial distribution indicated a pronounced preference for base composition at the synonymous third codon position, specifically showing that A/C were more prevalent than T/G. However, in the case of *ND1*, *ND4*, *ND4L*, and *ND5*, the points were concentrated within the fourth quadrant, implying that codon usage preferences adhered to the principle of T > A and G > C at the third codon position. Furthermore, four Gastrozonini species exhibited irregular distribution patterns, further demonstrating that the frequencies of AT and GC nucleotides were not utilized equally at the synonymous third codon position across the 13 mitochondrial genes. Therefore, these findings suggested that both mutational pressure and natural selection jointly influenced CUB in Dacinae species.

### 3.5. Gene Overlap and Intergenic Regions

In the mitogenome of *A. separata*, a total of 6 gene overlaps were identified, with overlap lengths spanning from 1 to 8 bp, culminating length of 27 bp. Comparatively, the mitogenomes of *A. quadrivittata*, *G. parviseta*, and *P. vulgaris* exhibited 6, 7, and 6 gene overlaps, with similar size ranges, totaling 26 bp, 27 bp, and 26 bp, respectively (Appendix A). Moreover, the most extensive overlap region across these four mitogenomes was 8 bp, located at the junction between the *trnW* and *trnC*. Regarding intergenic regions, *A. separata* contained 20 intergenic spacers, with spacer lengths varying from 1 to 91 bp and a combined total length of 222 bp. In comparison, *A. quadrivittata*, *G. parviseta*, and *P. vulgaris* possessed 20, 19, and 20 intergenic spacers, with sizes ranging from 1 to 98 bp, 104 bp, and 90 bp, summing to total lengths of 222 bp, 228 bp, and 215 bp, respectively. The longest intergenic spacer identified was 104 bp within these four species, situated between *trnA* and *trnR*. Within the 46 Dacinae mitogenomes, gene overlaps were typically short, predominantly spanning from 1 to 8 bp. But the most extensive overlap observed was 1024 bp, occurring between *trnS2* and *nad1* in *Dacus* (*Didacus*) *ciliatus* (Dacini); it is worth noting that this sequence was unverified. Conversely, the lengths of intergenic regions exhibited considerable variability across these genomes, with the most extensive intergenic segment reaching up to 967 bp, located between *trnS2* and *trnL1* in *D.* (*D.*) *ciliatus*.

### 3.6. RNA Genes

In the four newly sequenced mitogenomes, the 22 tRNA genes were distinctly distributed throughout the entire mitogenome. The cumulative lengths of these tRNA were 1470 bp in *A. separata*, 1474 bp in *A. quadrivittata*, 1469 bp in *G. parviseta*, and 1473 bp in *P. vulgaris*, all exhibiting a substantial AT nucleotide bias reaching 77.2%, 78.7%, 76.1%, and 76% in these respective species (Appendix A). Four species displayed a negative AT-skew coupled with a positive GC-skew across their tRNA genes. Furthermore, extending this analysis to a broader dataset of 46 Dacinae mitogenomes demonstrated a significant AT bias within tRNAs, ranging from 71.5% in Dacini, *B*. (*T*.) *tsuneonis* to 78.7% in Gastrozonini, *A. quadrivittata*, averaging approximately 75.5%. This AT content was higher than that observed in PCGs, yet lower than that in rRNAs and CR (Figure 4). Additionally, all tRNA genes exhibited positive GC-skew values, while the majority of AT-skew values were positive. A comprehensive comparison of the predicted secondary structures of all 22 tRNAs across 46 Dacinae mitochondrial genomes revealed the presence of mismatch pairs within the stems of several tRNAs. The most prevalent mismatch pattern was the GU, followed by other non-canonical pairings such as UU, CU, and AA. Notably, all anticodon loops within the Dacinae mitogenomes were highly conserved and identical, measuring 7 bp in length, whereas the lengths of DHU and TΨC loops exhibited greater variability (Appendix A).

The two rRNA genes, namely *rrnL* (large rRNA) and *rrnS* (small rRNA), were identified in the four newly sequenced mitogenomes. Specifically, the *rrnL* was consistently located between the *tRNA-Leu1* and *tRNA-Val* across all four species, with lengths of 1334 bp in *A. separata*, 1339 bp in *A. quadrivittata*, 1333 bp in *G. parviseta*, and 1334 bp in *P. vulgaris*. Conversely, the *rrnS* was positioned between the *tRNA-Val* and CR, measuring 797 bp in *A. separata*, 793 bp in *A. quadrivittata*, 797 bp in *G. parviseta*, and 788 bp in *P. vulgaris* (Appendix A). These four rRNA genes demonstrated a pronounced AT bias, with overall AT content reaching up to 80% in *A. separata*, 81.7% in *A. quadrivittata*, 79% in *G. parviseta*, and 79.3% in *P. vulgaris*. Additionally, these four rRNAs possessed negative AT-skew values and positive GC-skew values (Appendix A). A comparative examination of mitogenomes across 46 Dacinae species demonstrated a remarkable AT bias within the rRNA, with AT content ranging from 73.4% (Dacini, *B*. (*T*.) *tsuneonis*,) to as high as 81.9% (Ceratitidini, *N. asiatica*), averaging approximately 78.37%, which surpassed that of PCGs and tRNAs (Figure 4). Apart from *Zeugodacus* (*Zeugodacus*) *caudatus*, the AT-skew values across all species remained negative, while GC-skew values persisted as positive in their rRNA sequences.

### 3.7. A+T-Control Region

In the mitogenomes of Dacinae, the putative A+T-control region was typically situated between the *rrnS* and *trnI* (Figure 1). The length of this control region varied among four species, measuring 1623 bp in *A. separata*, 1132 bp in *A. quadrivittata*, 1717 bp in *G. parviseta*, and 1636 bp in *P. vulgaris*, with corresponding AT content reaching 82.1%, 79.8%, 79.3%, and 74.7%, respectively (Appendix A). The calculations of AT-skew and GC-skew exhibited high variability within the CR across the 46 Dacinae mitogenomes. Missing data or incomplete measurements for the CR could have contributed to the apparent length variations. The control region demonstrated the highest AT content when relative to the overall mitochondrial genome, PCGs, and RNAs, averaging up to 86.23% (Figure 4).

### 3.8. Heterogeneity of Base Composition

To ascertain dataset heterogeneity and the need for a heterogeneous model, we employed AliGROOVE to assess sequence similarity and alignment ambiguity in nucleotide and amino acid matrices. The results demonstrated that pairwise sequence comparisons across the six datasets generally exhibited relatively high similarity scores and demonstrated low levels of heterogeneity in sequence divergence (Figure 11). However, the heterogeneity associated with the PCGs was notably higher than that observed in the AA dataset, with the third codon position of PCGs showing the highest degree of heterogeneity, characterized by the lowest pairwise sequence similarity scores. Conversely, the AA dataset demonstrated comparatively higher similarity. This elevated divergence in the PCG3 dataset was primarily attributed to the rapid evolutionary rate at the third codon position, promoting substitution saturation and leading to markedly greater heterogeneity compared to analyses of the first and second codon positions. As a consequence, a dataset of concatenated PCGs excluding the third codon position can effectively mitigate the degree of heterogeneity. Furthermore, our analyses identified the following taxa: *Bactrocera* (*Tetradacus*) *minax*, *B.* (*T.*) *tsuneonis*, *Dacus* (*Didacus*) *ciliates*, and *Zeugodacus* (*Zeugodacus*) *caudatus* with relatively pronounced levels of sequence heterogeneity. These taxa may pose challenges for accurate phylogenetic placement, potentially leading to unstable or incorrect positioning within phylogenetic trees.

### 3.9. Phylogenetic Reconstruction Using Site-Homogenous Models

Phylogenetic reconstructions employing both Bayesian inference (BI) and Maximum Likelihood (ML) methods were conducted based on across five diverse datasets (PCG123, PCG123R, PCG12, PCG12R, and AA) under the site-homogenous models. These results consistently produced ten phylogenetic trees with nearly identical topologies and predominantly well-supported branches, while a few branches exhibited minor instability (Figure 12 and Appendix A). Crucially, the defined putative ingroup Dacinae, encompassing three tribes, was robustly resolved as monophyletic relative to four subfamilies of Tephritidae, including Tachiniscinae, Tephritinae, Phytalmiinae, and Trypetinae, demonstrating a highly reliable phylogenetic topology for Dacinae (BS ≥ 80, PP = 1). Additionally, this subfamily robustly recovered a clade composed of Gastrozonini and Ceratitidini as the sister group to Dacini across all analyses. Also, phylogenetic analysis confirmed the monophyly of these three tribes and produced robust estimates for their relationships, with unwavering clade support in BI and ML trees (BS = 100, PP = 1).

Although the three tribes were consistently found to be as the monophyletic group in our analyses, the generic relationships within the Ceratitidini and Dacini remained unstable across different datasets. Specifically, within the tribe Ceratitidini, *Neoceratitis asiatica* formed a single clade and was sister to *Ceratitis* across the PCG12, PCG12R, PCG123, and PCG123R datasets with a robust support value at node (BS = 100, PP = 1), while *N. asiatica* was embedded into the representative species of *Ceratitis*, resulting in a non-monophyletic genus in the AA dataset. Furthermore, *Bactrocera* and (*Dacus* + *Zeugodacus*) were recovered as sister groups with high branch support for PCG123R-BI + ML analyses (BS = 100, PP = 1), affirming the monophyly of these three genera. In contrast, the representative species of *Zeugodacus* and *Dacus* were grouped together based on the AA-ML, PCG12R-BI + ML, and PCG123-BI + ML analyses with the exception of *Zeugodacus* (*Parasinodacus*) *cilifer*. Similarly, for the AA-BI and PCG12-ML analyses, *Z.* (*P.*) *cilifer* and two *Bactrocera* species formed a clade with low-to-moderate branch support (BS = 24, PP = 0.82), whereas in the PCG12-BI analysis, they formed a separate clade (Appendix A). Consequently, the phylogenetic resolution of the two genera *Bactrocera and Zeugodacus* was not well-defined and not confirmed to be monophyletic for certain datasets, and the phylogenetic placement of individual species within the genus has changed.

### 3.10. Phylogenetic Reconstruction Using Site-Heterogeneous Model

Phylogenetic analyses were performed under the site-heterogeneous mixture model CAT + GTR using the Bayesian inference method across five different datasets, specifically including PCG123, PCG123R, PCG12R, PCG12, and AA. Apart from the PCG123 and PCG123R datasets, the results consistently supported the monophyly of the subfamily Dacinae and yielded slightly divergent topologies for tribal relationships, with most nodes being supported by moderate-to-high posterior probability values (Figure 13 and Appendix A). Importantly, the three representative tribes examined were robustly recovered as monophyletic with strongly supported posterior probabilities. Within this subfamily, the clade comprising Ceratitidini and Gastrozonini was consistently sister to the Dacini across the AA, PCG12, and PCG12R datasets. Conversely, in the PCG123 and PCG123R datasets, Gastrozonini, in conjunction with the representative species of the outgroup, were grouped together (Appendix A). Regarding the tribe Ceratitidini, *N. asiatica* clustered closely with certain species of *Ceratitis* or was embedded into the *Ceratitis* representatives across all datasets. Furthermore, the few species of *Bactrocera* and *Zeugodacus* formed a distinct clade as the sister group to the remaining species of *Zeugodacus* and *Dacus* within Dacini across all analyses, giving rise to the monophyly of the two genera *Bactrocera* and *Zeugodacus*, which has not been verified. Overall, the phylogenetic placements of individual species within the two genera *Bactrocera* and *Zeugodacus* were unstable, leading to fluctuating intergeneric relationships. Despite the convergence indicated by the maxdiff values across the five datasets, several unstable branches remained poorly resolved and did not achieve satisfactory resolution.

## 4. Discussion

Currently, gene rearrangement has not been observed in the mitochondrial genomes of Dacinae species, but tRNA gene duplication have been documented in prior studies [33]. The presence of a truncated stop codon, such as the T–, appears to be a relatively common phenomenon and is typically converted into a complete stop codon like TAA through post-transcriptional modifications during the process of mRNA maturation [68]. This pattern aligns with observations in other true fruit flies, as reported in recent research [32,52]. Furthermore, with the exception of *trnS1*, all inferred tRNAs can be folded into the canonical cloverleaf secondary structure, wherein the dihydrouridine (DHU) arm of *trnS1* is substituted with a single loop, as was also observed in other Tephritoidea species [69]. The atypical secondary structure of *trnS1* may arise from transcriptional or post-transcriptional anomalies, which are thought to have originated early in evolutionary history and occur frequently across the mitochondrial genomes of Metazoa [26,70]. Additionally, the A+T-rich region plays a crucial role in the regulation of mitochondrial gene replication and transcription processes [29,71].

In the present study, the high support values at the nodes, the stability of the branch structures, the minimal occurrence of phylogenetic errors, and the optimal phylogenetic hypothesis suggested that the results obtained from site-homogeneous models, in conjunction with the currently accepted phylogeny, were superior to those derived from the heterogeneous model based on the same dataset. Sequence divergence analyses certified the low heterogeneity with overall datasets, and the CAT + GTR model may be less appropriate for these small datasets due to limited sequence length and information [72]. These indicated that homogeneous models may generally be more advantageous for phylogenetic reconstruction within this group.

Building upon the findings of previous phylogenetic studies [1,23,24,49], this research reaffirmed the monophyly of Dacinae across different datasets. However, it also challenged prior studies that considered this subfamily to be paraphyletic [32,33,34]. The subfamily Dacinae, which encompasses three primary tribes including Gastrozonini, Dacini, and Ceratitidini, was selected as representative species, and their monophyly was consistently recovered. Specifically, Gastrozonini and Ceratitidini were regarded as the early-diverging lineages and resolved as the sister group to Dacini, which was also consistent with prior phylogenetic analyses [24,73]. The stability of the phylogenetic status among these three tribes was robust across all analyses, which yield largely congruent topologies and well-supported values for most clades, with the exception of few unstable branches under both site-homogenous and site-heterogeneous mixture models (Figure 12 and Figure 13 and Appendix A). However, with respect to the tribe Ceratitidini, our analyses revealed that the placement of *N. asiatica*, which was embedded into other *Ceratitis* representatives, was inconsistent with the currently established phylogenetic relationships. This discrepancy has contributed to the unresolved monophyly of *Ceratitis* across certain datasets, highlighting the need for further investigation. While the monophyly of Dacini was strongly proved, the phylogenetic positions and relationships of specific species within the genera *Bactrocera* and *Zeugodacus* remain poorly defined and contentious, which is in accordance with previous phylogenetic investigations [74,75]. Especially in the PCG12 and AA datasets, different methods for constructing phylogenetic trees affected the phylogenetic status of few species, which suggested that these two datasets were more sensitive to the tree-constructing methods. As a result, only a few species exhibited shifts in their phylogenetic positions, resulting in different topologies for certain clades. To improve resolution and achieve a more definitive understanding of these relationships, future research should aim to expand the current dataset through broader taxon sampling, and a comprehensive approach combining molecular and morphological evidence can provide a more holistic understanding of evolutionary relationships and support more accurate taxonomic classifications.

## 5. Conclusions

In this study, we sequenced and analyzed four complete mitogenomes for the first time, specifically *Acroceratitis separata*, *Acrotaeniostola quadrivittata*, *Gastrozona parviseta*, and *Paragastrozona vulgaris*, which represent Dacinae species. These four mitogenomes contained typical 37 mitochondrial genes (13 PCGs, 22 tRNAs, and two rRNAs) and an A+T-control region, exhibiting the notable size variation ranging from 16,112 bp to 16,691 bp. The observed size variation primarily correlated with mutability in the length of the A+T-rich region among Dacinae mitogenomes. A remarkedly high AT content was consistently observed across multiple genomic regions, including PCGs, RNAs, and CR. The Dacinae mitogenomes were highly conserved in genome size and gene order, base content and composition, PCGs and codon usage, and the secondary structure of tRNAs. Additionally, pairwise similarity comparisons of PCGs indicated that *ND2*, *ATP8*, *ND3*, *ND5*, and *ND6* were relatively highly variable and less conserved, while *COI*, *COIII*, and *CYTB* showed a lower degree of variation. We calculated several indices of CUB and suggested that both mutational pressure and natural selection jointly influenced codon usage of Dacinae species. Subsequently, we reconstructed the phylogenetic relationships within Dacinae based on Bayesian inference and Maximum Likelihood methods, utilizing site-homogeneous models, which outperformed the site-heterogeneous mixture model in terms of phylogenetic resolution. Despite a few species demonstrating unstable placements, the phylogenetic relationships among the three representative tribes were recovered as monophyletic groups, with the topology represented as ((Ceratitidini + Gastrozonini) + Dacini), and most branches displaying moderate-to-high support values. In conclusion, to achieve a more refined understanding of the varied clades and contentious relationships within this group, it is imperative to incorporate an increasing number of molecular data with broader generic representation. This study significantly contributes to the existing mitogenomic data, enhancing our comprehension of the evolutionary trajectories and higher-level phylogenetic relationships within the Dacinae.

## Figures and Tables

**Figure 1 animals-15-03301-f001:**
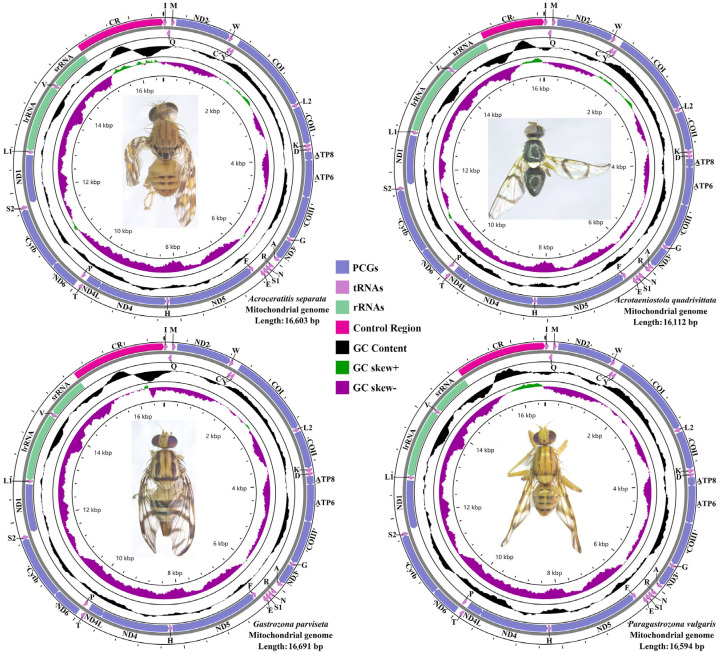
Circular map of the complete mitochondrial genome of *Acroceratitis separata*, *Acrotaeniostola quadrivittata*, *Gastrozona parviseta*, and *Paragastrozona vulgaris*.

**Figure 2 animals-15-03301-f002:**
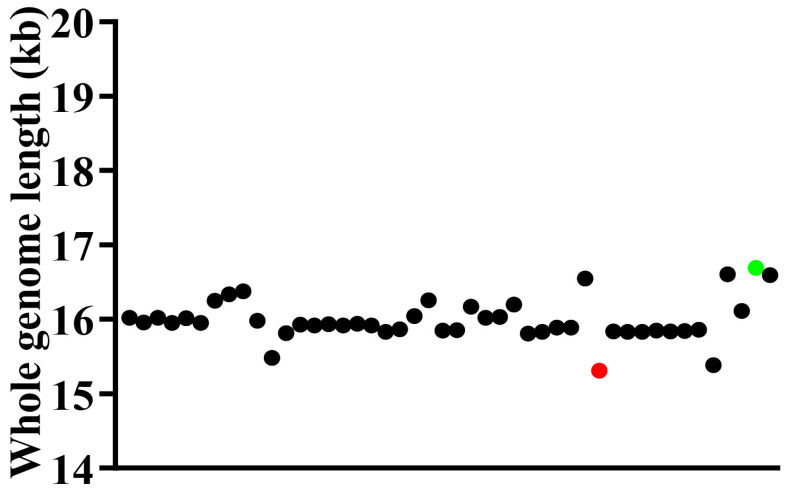
The scatterplot of 46 Dacinae mitogenomic sizes with a red dot and green dot, respectively, representing *Zeugodacus* (*Zeugodacus*) *caudatus* and *Gastrozona parviseta*.

**Figure 3 animals-15-03301-f003:**
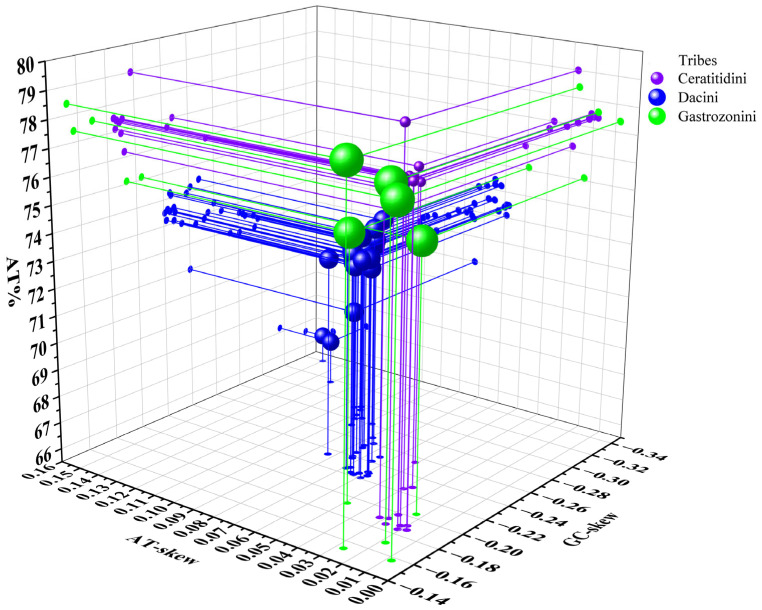
Three-dimensional scatter plots of the AT-skew, GC-skew, and AT content of 46 Dacinae mitochondrial genomes. Balls of different colors correspond to different tribes.

**Figure 4 animals-15-03301-f004:**
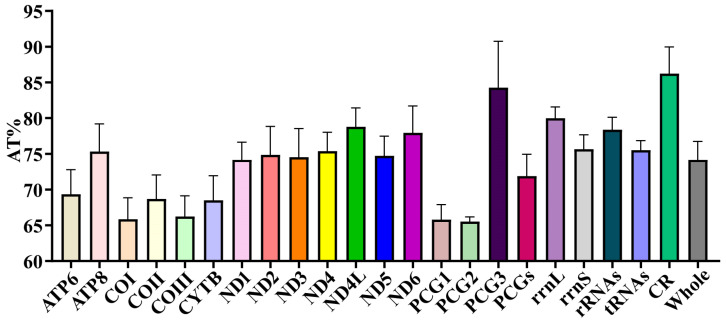
The average AT content value for each component within 46 Dacinae mitogenomes. Error bars represent standard deviations from data of multiple species. The abbreviations PCG1, PCG2, and PCG3, respectively, represent the first, second, and third codon position of PCGs.

**Figure 5 animals-15-03301-f005:**
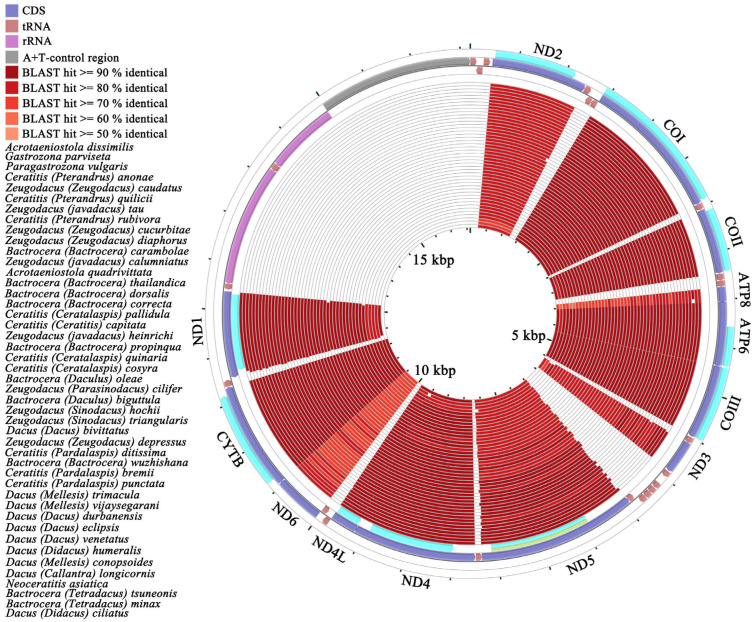
A graphical map of the BLASTP result showing the PCG similarities between the *Acroceratitis separata* and that of other Dacinae mitogenomes. Gene regions and BLAST identities are shown from outside to inside. The CCT arranges the BLASTP result such that the sequence most similar to the reference sequence (*A. separata*) is placed closer to the outer edge of the map.

**Figure 6 animals-15-03301-f006:**
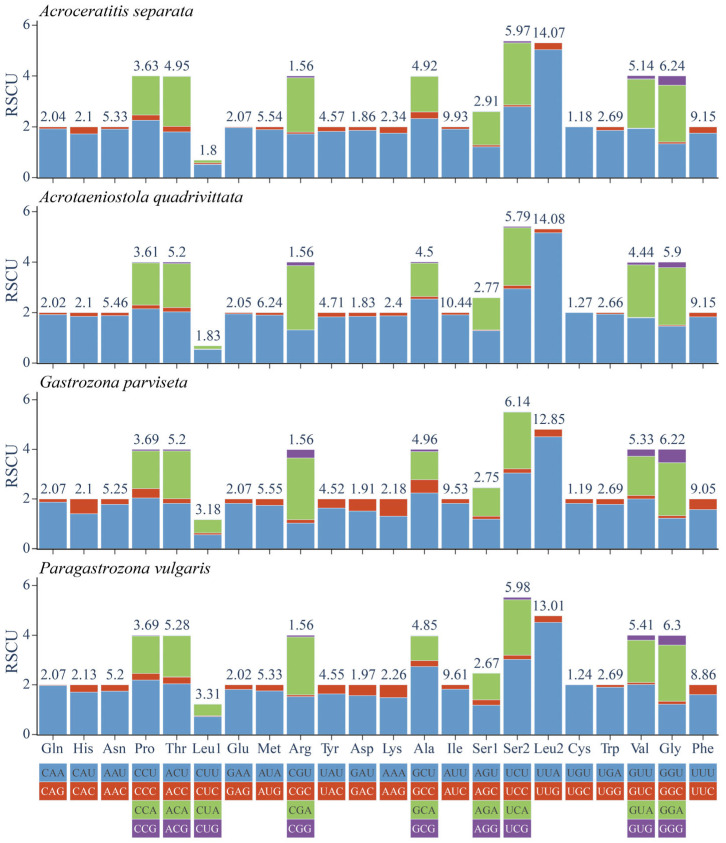
Relative synonymous codon usage (RSCU) in PCGs of four dacine mitogenomes. The *Y*-axis scale represents the RSCU value and the numerical value above the bars graph indicates the AA ratio.

**Figure 7 animals-15-03301-f007:**
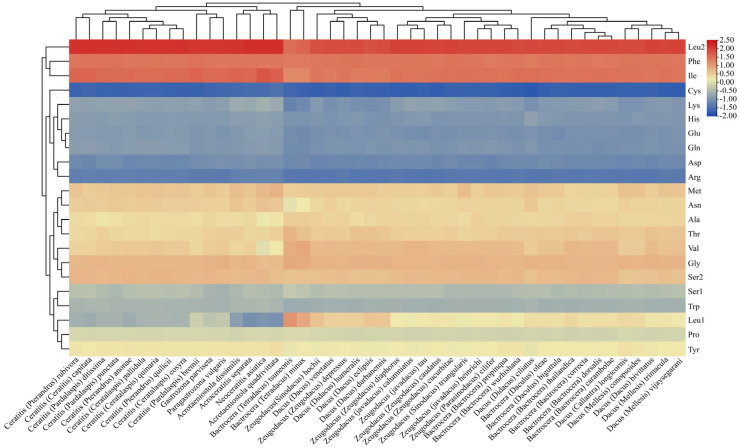
Hierarchical clustering analysis for the heatmap showing AA usage ratio of 46 mitogenome species within Dacinae.

**Figure 8 animals-15-03301-f008:**
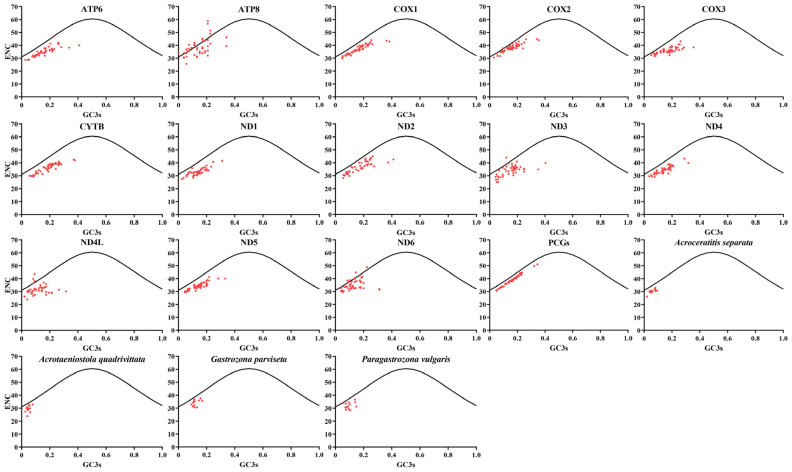
ENC-GC3s plot analysis of all 13 PCGs and each PCG within 46 Dacinae mitogenomes, as well as 13 PCGs from four dacine mitogenomes, respectively.

**Figure 9 animals-15-03301-f009:**
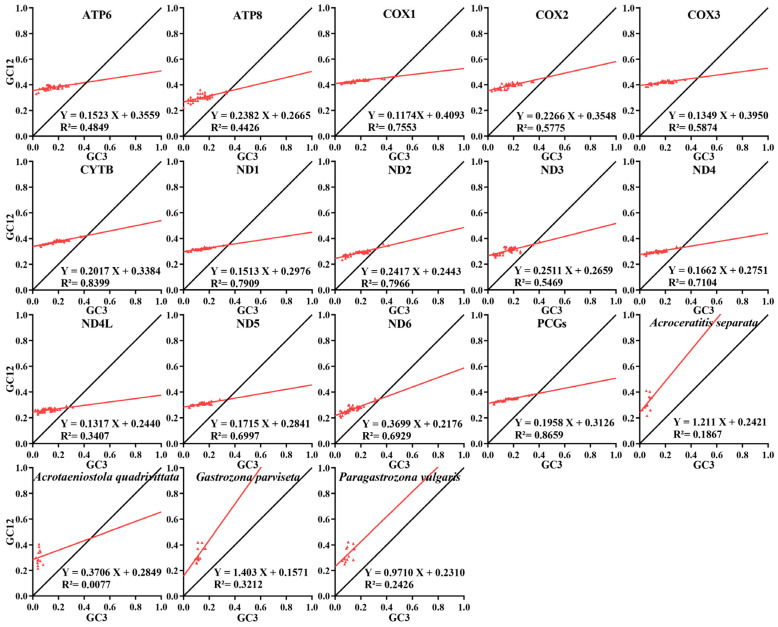
Neutral plot analysis of all 13 PCGs and each PCG within 46 Dacinae mitogenomes, as well as 13 PCGs from four dacine mitogenomes, respectively.

**Figure 10 animals-15-03301-f010:**
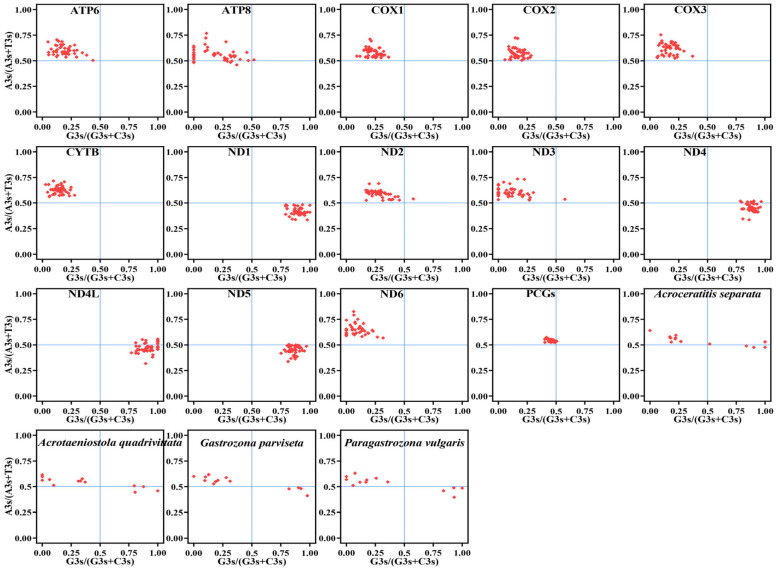
Analysis of PR2 bias plot including all 13 PCGs and each PCG within 46 Dacinae mitogenomes, as well as 13 PCGs from four dacine mitogenomes, respectively.

**Figure 11 animals-15-03301-f011:**
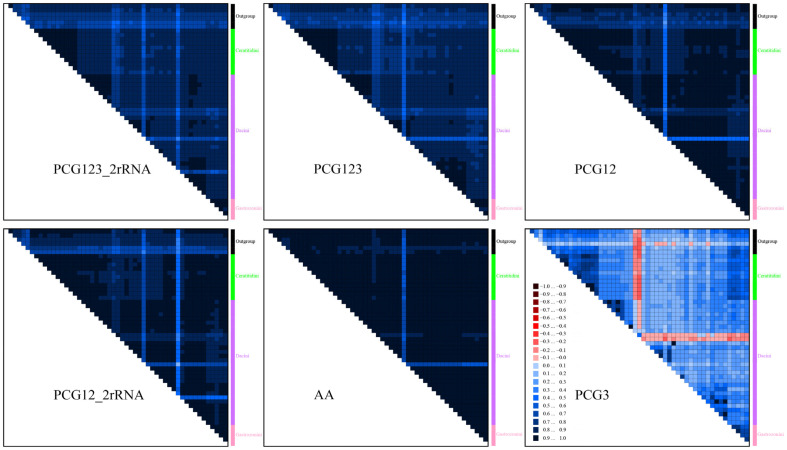
The heterogeneity of the sequence composition of the mitochondrial genomes in different datasets. The mean similarity score between sequences is represented by a colored square based on AliGROOVE scores ranging from −1, indicating full random similarity (heterogeneity, red coloring), to +1, indicating nonrandom similarity (blue coloring).

**Figure 12 animals-15-03301-f012:**
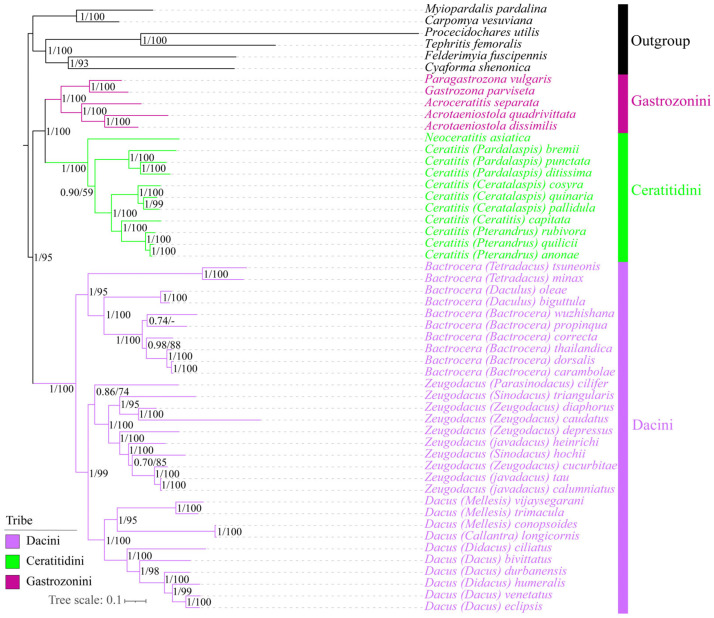
Phylogenetic tree of Dacinae inferred from PCG123R-BI + ML datasets using Bayesian inference and Maximum Likelihood methods. Numbers on branches are posterior probabilities and bootstrap support values.

**Figure 13 animals-15-03301-f013:**
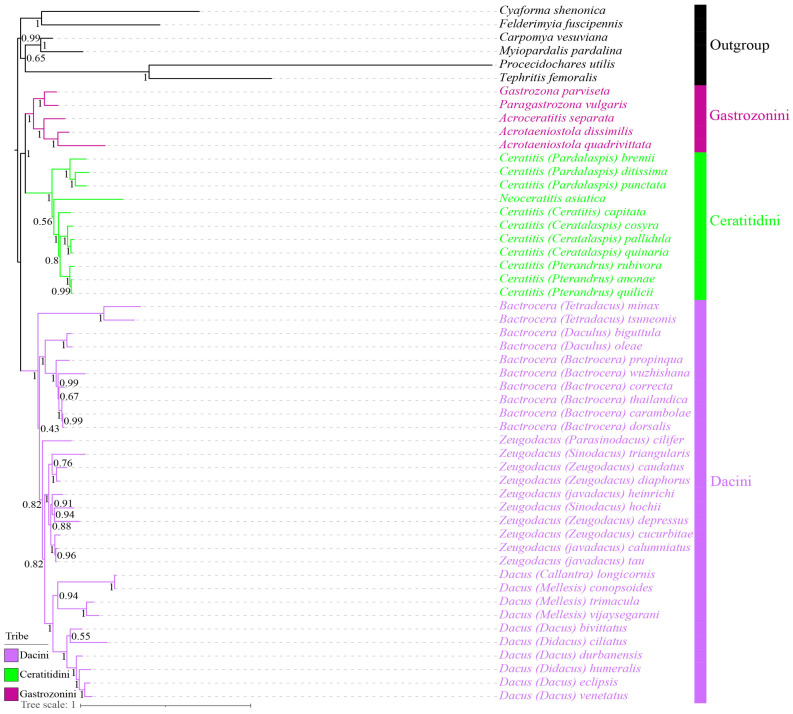
Phylogeny of Dacinae inferred from the AA dataset using PhyloBayes analysis under the site-heterogeneous mixture model CAT + GTR. Supports at nodes are Bayesian posterior probabilities.

## Data Availability

The data presented in this study were submitted to the NCBI databases at https://www.ncbi.nlm.nih.gov/ (accessed on 10 July 2025) and can be accessed with accession numbers PV920620 (*Acroceratitis separata*), PV920621 (*Acrotaeniostola quadrivittata*), PV920622 (*Gastrozona parviseta*), and PV920623 (*Paragastrozona vulgaris*).

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
