# Peer review of "Characterization of the Complete Mitogenomes of Four Dacinae Species (Diptera: Tephritidae) with Phylogenetic Analysis"

_animals, 2025, doi:10.3390/ani15223301_

Round 1
Reviewer 1 Report
Comments and Suggestions for Authors
Please see the attachment.

Reviewer 2 Report
Comments and Suggestions for Authors
I am grateful to the authors for this interesting manuscript. This work fully complies with the scope of the journal Animals.
The work is based on receptive material, statistically analyzed, and contains unambiguous results.
Overall, I would like to note the scientific novelty of this work. The authors sequenced the complete mitogenomes of four representatives of the subfamily Dacinae. They clarified the position and evolutionary relationships within the group, and resolved some relationships within Dacinae. The results obtained are of fundamental importance for the general entomology and taxonomy of dipteran insects.
I recommend to the Editor accept the manuscript as is.
Reviewer 3 Report
Comments and Suggestions for Authors
This study reported the mitogenomes of four fruit fly species. These mitogenome sequences also use to infer phylogenetic relationships with several previously recorded fruit fly species. In my opinion, the information present in this study is interesting and I have comments and suggestions for further improvement of the manuscript.
- I think the abstract lacking the key findings of this study. The current version focus mainly on methodology. I suggest that this part should be revised and added the significant finding such as characteristics of mitogenome of these four fruit fly species, are they similar or different from previous recorded from other fruit fly species, positions of these four species in the phylogenetic analyses, etc.
- L30, what is PCGs? It should be write in full for the first mention.
- L616-618, I agree that expanding dataset by broader taxon sampling can help to improve phylogenetic analyses. However, I also wondering whether that the question of morphological taxonomic placement of the species can also discuss?
- Conclusion, this section mostly repeats the result section. I suggest that it should present the key finding of the present study and provide the suggestion for further research on such topic.
